# The effect of maternal care on gene expression and DNA methylation in a subsocial bee

Samuel V. Arsenault [1], Brendan G. Hunt [1] & Sandra M. Rehan[2]

Developmental plasticity describes the influence of environmental factors on phenotypic variation. An important mediator of developmental plasticity in many animals is parental care. Here, we examine the consequences of maternal care on offspring after the initial mass provisioning of brood in the small carpenter bee, *Ceratina calcarata*. Removal of the mother during larval development leads to increased aggression and avoidance in adulthood. This corresponds with changes in expression of over one thousand genes, alternative splicing of hundreds of genes, and significant changes to DNA methylation. We identify genes related to metabolic and neuronal functions that may influence developmental plasticity and aggression. We observe no genome-wide association between differential DNA methylation and differential gene expression or splicing, though indirect relationships may exist between these factors. Our results provide insight into the gene regulatory context of DNA methylation in insects and the molecular avenues through which variation in maternal care influences developmental plasticity.

[1] Department of Entomology, University of Georgia, Athens, GA 30602, USA. [2] Department of Biological Sciences, University of New Hampshire, Durham, NH 03824, USA. Correspondence and requests for materials should be addressed to B.G.H. (email: huntbg@uga.edu) or to S.M.R. (email: sandra.rehan@unh.edu)

Developmental plasticity is the capacity of an organism to produce multiple stable phenotypes from a single genotype given differing environmental conditions[1]. Examples include changes in stress responses[2] and caste differentiation[3]. Social experiences occurring early in development, such as maternal care, are particularly influential on the developing brain[4]. Variation in maternal care has been linked to variation in stress responses of offspring in rats;[5] this effect is reversible by cross-fostering and is mediated by changes in epigenetic information, such as DNA methylation[5,6]. In bees, variation in the social environment during larval development also alters the expression of genes that regulate behavioral and endocrine responses to stress[7]. Moreover, a recent comparison of vertebrate and honey bee responses to social challenge suggests that fundamental commonalities exist in the molecular mediation of variation in social responses across diverse animal taxa[8]. We were curious as to whether variation in parental care in insects results in long-term effects on offspring behavior, as in mammals, and whether common molecular mediators may exist for such behavioral convergence.

The subsocial small carpenter bee *Ceratina calcarata* is an ideal system for addressing questions regarding the effect of developmental environment and parental care on behavior and gene regulation. Unlike its solitary relatives, *C. calcarata* exhibits maternal care behaviors that consist of mass provisioning prior to oviposition, offspring grooming, and nest defense[9]. Mothers of this species are long lived and nest loyal, producing only a single nest in their lifetime and surviving to interact with adult offspring[10]. Adult offspring are typically non-aggressive in the presence of mothers and exhibit mutual tolerance prior to overwintering[11,12]. All of these behaviors take place in the isolated confines of a narrow, compartmentalized twig. Here, we explore the effects of maternal care on offspring development after the initial mass provisioning of brood.

We compare phenotypic and gene regulatory differences between maternally cared for and orphaned individuals, assay aggression and avoidance behaviors, perform RNA-sequencing to measure changes in transcription, and perform bisulfite sequencing to quantify changes in DNA methylation. Similar studies, performed in mammals, have shown that varying maternal care conditions can have effects on DNA methylation and gene expression that persist throughout the lives of offspring[2,5]. However, the gene regulatory consequences of DNA methylation are more poorly understood in insects than in mammals, and it is presently unknown whether DNA methylation mediates potential long-term effects of variation in parental care in *C. calcarata* or other subsocial insects.

Previous work in *C. calcarata* has shown that its genome contains the requisite machinery to establish and maintain DNA methylation[13]. Additionally, the patterning of DNA methylation is similar to what has been observed in many hymenopteran insects, with almost exclusively intragenic DNA methylation, biased to constitutively expressed genes[14]. Two primary mechanistic explanations have been proposed to link gene body DNA methylation to the regulation of environmental effects on development in insects[15–17]. The first is that DNA methylation is involved in mediating changes in gene expression levels. The effects of DNA methylation on gene expression have been extensively studied in mammals since the 1970s[18]. It has long been thought that methylation in promoter regions leads to suppression of gene expression[18], though this idea has recently been challenged[19]. Despite its supposed suppressive role in the promoter region of genes, DNA methylation within the gene body is, overall, positively correlated with transcription in mammals[20] as well as insects[14]. This positive correlation could be explained by preferential targeting of DNA methylation to actively transcribed genes rather than a direct effect of DNA methylation on transcription. Indeed, the presence of gene body DNA methylation may actually reduce gene expression levels by impeding transcriptional elongation efficiency[15,21]. Thus, the relationship between gene body DNA methylation and the regulation of gene expression levels is complex and remains poorly understood. A second proposed mechanism by which gene body DNA methylation may affect gene function is through the regulation of alternative splicing[22]. It is known that DNA methylation affects transcription factor binding[23], which in gene bodies affects RNA Polymerase II processivity and the recognition of variably utilized splice sites[22]. While this phenomenon has not been well studied in invertebrates beyond genome-wide correlations, a few studies have supported a link between DNA methylation and alternative splicing in insects[24–27]. Either or both of these proposed mechanisms could feasibly link DNA methylation to the regulation of developmental plasticity.

The purpose of our study is twofold. First, we investigate the effects of variation in maternal care on offspring behavior and gene regulation, using the subsocial bee *C. calcarata* as a model. Second, we investigate whether stable differences in gene regulation observed between adults from distinct developmental environments are associated with, and potentially driven by, differences in DNA methylation.

We hypothesized that differences in maternal care may lead to changes in DNA methylation and consequently transcriptional differences in the form of differential gene expression or alternative splicing. We found significant changes in gene expression and splicing between individuals raised with a mother and those raised without, and comparatively small changes in DNA methylation. Additionally, we found very little overlap between the changes in DNA methylation and the changes in gene expression or splicing. We hypothesize that the role of DNA methylation is more subtle and nuanced than previously appreciated and may work in concert with other factors to yield context-specific, non-global effects on gene regulation.

## Results

**Orphan offspring exhibit increased aggression and avoidance.** To test the effect of maternal care on behavior, we performed circle tube assays measuring the behaviors of newly eclosed adults. Recently eclosed adult bees raised from nests with and without mothers were placed in circle tubes and aggressive and avoidant behaviors were scored (Fig. 1b). Bees raised in the absence of their mother showed a significant increase in aggression and avoidance when compared to bees raised in the presence of a mother (Mann–Whitney *U*-tests $p < 0.03$; Fig. 1c).

**Loss of maternal care leads to many transcriptional changes.** Maternal care leaves a distinct mark on *C. calcarata* offspring, lasting into adulthood. In association with the behavioral changes described previously, 1450 genes of the 11,068 tested (13.1%) showed significant differential expression between pooled offspring raised with and without a mother (Fig. 2b; FDR-corrected $p < 0.01$). Functional enrichment of Gene Ontology (GO) biological process annotations were observed among differentially expressed genes for terms related to oxidation–reduction reactions, chitin, starch, and sucrose metabolic processes, and neurological functions (ionotropic glutamate receptor signaling pathway) (Table 1, Supplementary Data 1). These pathways represent possible genetic modules involved in offspring response to maternal care. Given the importance of maternal care in *C. calcarata*, we hypothesized that genes developmentally regulated by maternal care may also be subject to positive selection within the *C. calcarata* lineage. We compared differentially expressed

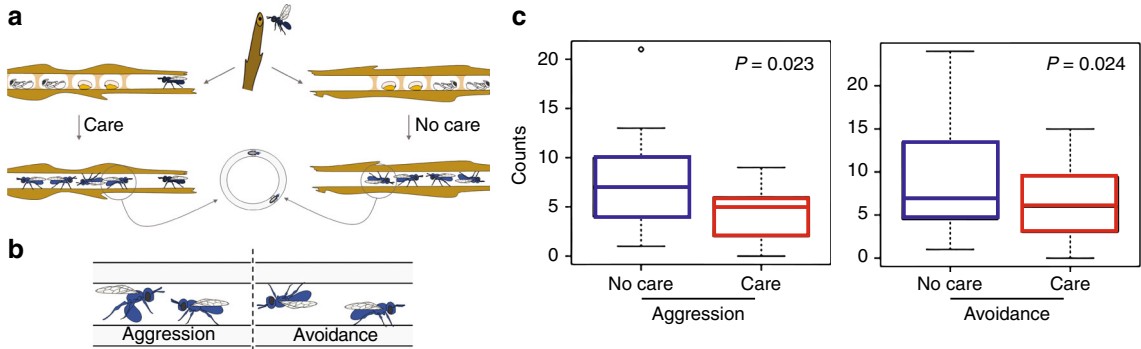

**Fig. 1** Behavioral response of adult offspring to early termination of maternal care. **a** Mothers establish nests in dead broken stems and produce all offspring in a linear hollow. Mothers typically remain on the nest to guard and groom immature offspring (care) or were removed from nests (no care). **b** Drawings of adult offspring engaging in aggression and avoidance behaviors in circle tube arenas. Illustrations in **a** and **b** courtesy of Wyatt Shell (used with permission). **c** Behaviors exhibited by offspring in circle tube assay in the presence and absence of maternal care (*p*-values are generated from two-tailed Mann–Whitney *U*-tests, $n = 52$; whiskers show observations within 1.5 × IQR of the lower and upper quartiles)

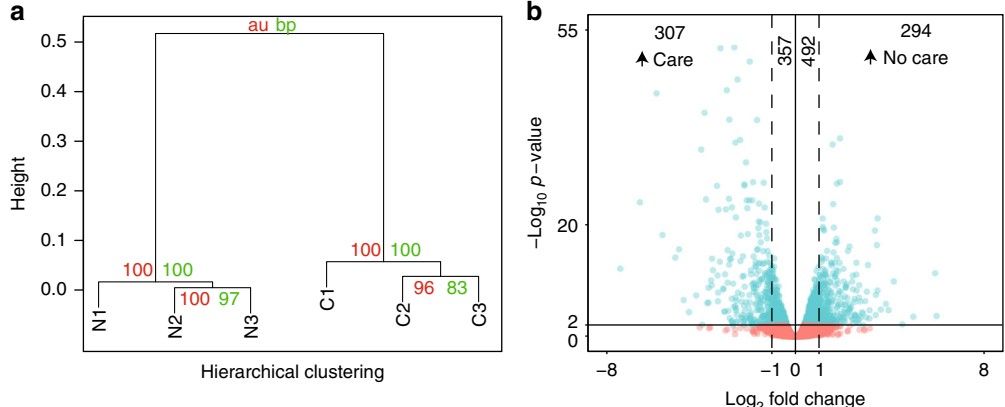

**Fig. 2** Loss of maternal care leads to extensive changes in gene expression. **a** Hierarchical clustering of RPKM values ($n = 11,068$ genes) using the "complete" method with correlation-based distances indicates that global gene expression levels cluster into "No Care" (N) and "Care" (C) groups, with high support values using both the approximately unbiased (au) and bootstrap probability (bp) statistics. **b** Volcano plot showing the genes considered differentially expressed (blue) and those not significantly differentially expressed (red). Genes that had more than a twofold difference in gene expression are shown outside of the dashed lines. The numbers of differentially expressed genes are indicated above

genes against a previously published list of genes showing signs of positive selection in the *C. calcarata* lineage, relative to other sequenced bee species, according to the branch-site test A in PAML[13,28]. We found no significant overlap between genes under positive selection and those that are differentially expressed when maternal care is removed (Chi-square test, $p = 0.72$; Supplementary Table 1).

In addition to identifying differentially expressed genes, we also searched for genes that showed signs of differential exon usage and other forms of alternative splicing, between the individuals raised with and without a mother. Production of different isoforms of various genes via alternative splicing is a well-established mechanism for generating developmentally plastic phenotypes[29]. We observed differential exon usage for 1090 exons spread across 373 genes (Supplementary Data 2). Two hundred and thirty-four genes were identified as alternatively spliced using rMATS, with these genes being largely mutually exclusive with the DEXseq-derived genes (Supplementary Figure 4). These results are indicative of changes in splicing regulation between the care and no-care individuals. Genes containing alternative splicing events were enriched for GO terms related to proteolysis, oxidation–reduction processes, and fatty acid biosynthetic processes to name a few (Supplementary Data 1; Fisher's Exact Test, *p*-value <0.01). Similar to results from the differentially

expressed genes, we found no significant overlap between alternatively spliced genes and genes that are putatively subject to positive selection in *C. calcarata* (Chi-square test, $p = 0.66$; Supplementary Table 2).

**Loss of maternal care subtly changes DNA methylation.** Using bisulfite sequencing, we measured DNA methylation at the single-nucleotide scale. For each of our samples of pooled individuals, we observed similar patterns to those found in the previously generated *C. calcarata* methylome[13] and methylomes from many other hymenopteran taxa[14,30–32] (Supplementary Figure 1, Supplementary Table 8). In particular, all of the newly generated *C. calcarata* methylomes show a bias towards exons at the 5′ end of genes (Supplementary Figure 1). Methylated genes (>5% mCG/CG) showed GO biological process functional enrichment for ribosome biogenesis, translation, and protein folding (Supplementary Data 1) and were less likely to be under positive selection when compared against unmethylated genes (Chi-square test, $p = 2.43\text{e-}07$; Supplementary Table 3). This corroborates previous results that DNA methylation is biased towards constitutively expressed genes, which tend to be subject to strong purifying selection[14,32,33].

**Table 1 Gene Ontology (GO) biological process term enrichment among genes differentially expressed between "care" and "no care" offspring (Fisher's exact test *p*-value <0.001)**

| GO accession | Term | Number of differentially expressed genes | Expected number of genes | *p*-Value |
|---|---|---|---|---|
| GO:0055114 | Oxidation–reduction process | 115 | 69.81 | 9.10E-10 |
| GO:0006030 | Chitin metabolic process | 29 | 7.84 | 3.90E-09 |
| GO:0005982 | Starch metabolic process | 9 | 1.96 | 1.60E-05 |
| GO:0005985 | Sucrose metabolic process | 9 | 1.96 | 1.60E-05 |
| GO:0034220 | Ion transmembrane transport | 43 | 31.81 | 9.30E-05 |
| GO:0005975 | Carbohydrate metabolic process | 55 | 29.55 | 0.00014 |
| GO:0035235 | Ionotropic glutamate receptor signaling pathway | 11 | 3.62 | 0.00032 |
| GO:0016042 | Lipid catabolic process | 11 | 7.84 | 0.00033 |
| GO:0006032 | Chitin catabolic process | 6 | 1.36 | 0.00064 |
| GO:0016998 | Cell wall macromolecule catabolic process | 6 | 1.36 | 0.00064 |

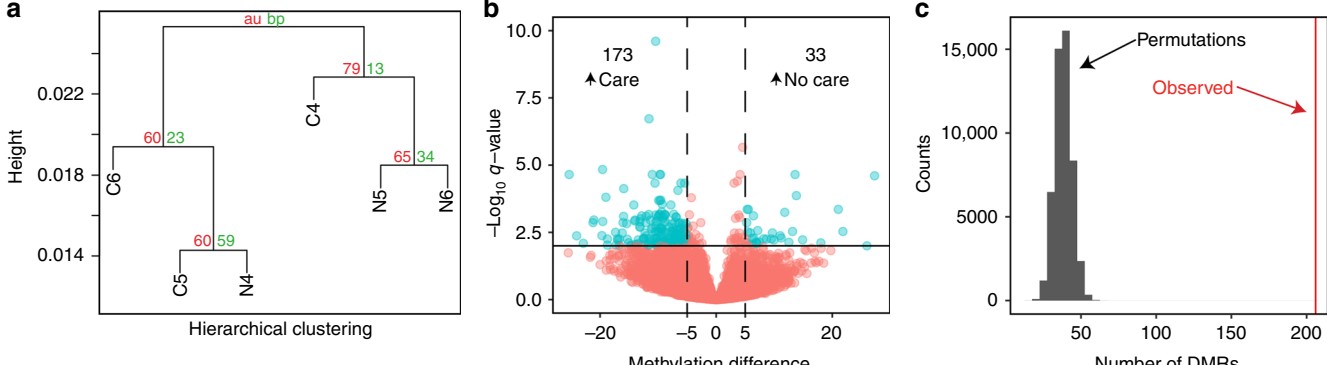

**Fig. 3** Loss of maternal care leads to localized, significant changes in DNA methylation. **a** Hierarchical clustering of gene level measures of DNA methylation ($n = 19,361$ genes) using the "complete" method with correlation-based distances reveals no global clustering by sample type (N, no care; C, care), with low support values using both the approximately unbiased (au) and bootstrap probability (bp) statistics. Thus, this tree can be considered one polytomy. Similar results are obtained when clustering only methylated genes. **b** Volcano plot showing the 200 bp regions considered differentially methylated (blue) and those not considered significant (red), as well as the thresholds used to define these categories (black lines). Methylation difference is given as a percentage. **c** Quadratic Assignment Protocol computed null distribution for the number of DMRs possibly occurring randomly using 50,000 randomized permutations and the associated number of experimental DMRs. The red line indicates the observed number of DMRs

Clustering analysis demonstrated that "no care" and "care" samples did not cluster by sample type based on global measures of DNA methylation (Fig. 3a). However, we also examined DNA methylation differences at a fine spatial scale using a differentially methylated region (DMR) approach. We searched for 200 bp windows throughout the genome that showed significant differential methylation between sample types (MethylKit, FDR-corrected *p*-value <0.01, >5% change in methylation). A total of 206 DMRs were identified between the individuals raised with maternal care and those raised without, spanning 238 gene models, with a substantial bias towards elevated DNA methylation in the individuals reared with a mother (84% of DMRs; Fig. 3b; Supplementary Data 2). The genes found to contain DMRs showed functional enrichment for translational elongation and metabolic processes (fructose, mannose, and superoxide) (Supplementary Data 1).

To further validate the presence of biologically meaningful DMRs, we utilized an adapted quadratic assignment method (QAM) to determine the number of DMRs we would expect to find if DNA methylation were distributed randomly with respect to sample type (e.g., if maternal care had no effect on DNA methylation). We found that the number of DMRs produced by proper sample assignment in our analysis (206) was substantially higher than the highest number of DMRs generated by QAM permutations of our data (65; mean = 38.2; Fig. 3c). This further

supported our finding that loss of maternal care itself led to significant, albeit localized, changes in DNA methylation. Additionally, it allows us to estimate how likely any of our individual DMRs occurred by chance (18.5%). We did not find significant overlap between differentially methylated genes and genes subject to positive selection (Chi-square test, $p = 0.54$; Supplementary Table 4).

**Independent variation in transcription and DNA methylation.** After identifying genes that have undergone changes in expression, splicing, or DNA methylation, we compared the relationship between these processes by characterizing overlap among gene sets (Fig. 4a). As observed previously in *C. calcarata* and other hymenopterans[13,32], we found a significant correlation between genic methylation levels and gene expression levels when averaged across all samples (Spearman $\rho = 0.24$, $p < 0.0001$). However, we found no evidence of greater overlap than expected by chance between genes containing DMRs and differentially expressed genes (Chi-square test, $p = 0.31$; Supplementary Table 5). Similarly, there was no significant overlap between genes containing DMRs and genes containing alternative splicing events (Chi-square test, $p = 0.4108$; Supplementary Table 6). Taken together, these findings indicate that there is not a consistent, genome-wide association between differential DNA

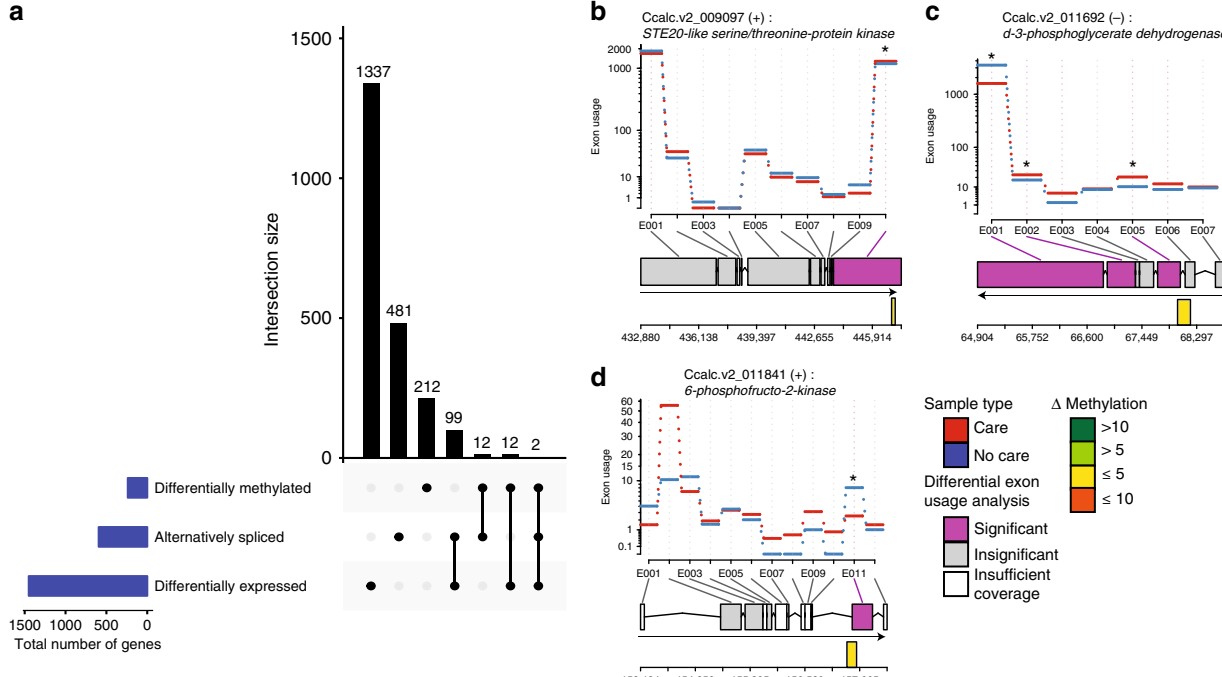

**Fig. 4** Overlap among differentially expressed, alternatively spliced, and differentially methylated genes. **a** UpSet plot illustrating the intersection between genes containing differentially methylated regions called by MethylKit, alternatively spliced genes called using DEXseq and rMATS, and differentially expressed genes computed using edgeR. The nature of a given intersection is indicated by the dots below the bar plot. For instance, the 99 genes in the fourth column are alternatively spliced and differentially expressed but not differentially methylated. **b–d** DEXseq plots showcasing the instances where a DMR directly overlapped a differentially utilized exon. The top panel shows average exon usage for each treatment group. The gene model is represented below each exon usage plot with purple exons indicating statistical support for differential usage (FDR-corrected *p*-value <0.01). Direction of transcription is indicated by an arrow and DMRs are overlaid beneath the gene model with colors indicating the change in methylation level

methylation and alternative splicing or differential expression in our data.

To further examine the potential relationship between DNA methylation and alternative splicing, we next investigated the possibility of spatial concordance between differential DNA methylation and alternative splicing by looking for direct overlap of 200 bp DMRs and various types of splicing events (Supplementary Figures 2 and 3). Among the eight genes that were both differentially methylated and alternatively spliced in our data, three had a DMR directly overlapping a differentially utilized exon (Fig. 4b–d, Supplementary Figure 2) while none of the alternative splicing events discovered using rMATS overlapped directly with a DMR (Supplementary Figure 3). These genes exhibit homology to genes encoding a STE20-like serine/threonine-protein kinase (SLK; Ccalc.v2_009097, XP_017892454), a D-3-phosphoglycerate dehydrogenase (PHGDH; Ccalc.v2_011692, XP_017888762), and a 6-phosphofructo-2-kinase (PFKFB1; Ccalc.v2_011841, XP_017879076). Each of these represents a candidate for localized involvement of DNA methylation in gene regulation associated with maternal care in *C. calcarata*.

## Discussion

In this study, we aimed to identify behavioral and molecular changes influenced by maternal care in developing *C. calcarata*. First, we quantified behavioral changes in newly eclosed adult offspring using circle tube assays. Then we measured changes in gene expression and alternative splicing using RNA-sequencing. Finally, we characterized DMRs in the genome using bisulfite sequencing. Taken together, these data point to a complex interplay of mechanisms underlying the effects of maternal care on developing offspring.

We observed differences in aggression and avoidance behaviors in *C. calcarata* offspring reared with and without maternal care. This pattern is reminiscent of patterns observed when maternal care varies in mammalian systems[34] and is consistent with effects of colony disturbance in honey bees[35]. Adaptive and non-adaptive explanations exist for these results. For example, divergent behaviors observed between bees reared with and without maternal care may simply reflect non-adaptive effects of a sub-optimal developmental environment. However, plasticity in aggression may also be influenced by natural selection if maternal death occurs regularly in the wild or variation in maternal care is prevalent. It is possible that bees reared in the absence of adult kin may incur fitness benefits from increased aggression and avoidance behaviors relative to bees that benefit directly from cooperative defense and care by kin as adults[12].

To further explore the relationship between maternal care and offspring development, we performed RNA-sequencing on the heads of newly eclosed bees that had been reared with or without a mother. Among the differentially expressed genes, we found enrichment of functional annotations related to metabolic processes, including oxidation–reduction reactions, and neurological processes, among others. These categories point to potential mechanisms for the increased aggression and avoidance behavior that we observed in offspring reared without maternal care.

Many of the differentially expressed genes we identified had functions relating to various forms of carbohydrate metabolism. While this is a very general functional category, variation in behavior has previously been linked to variation in carbohydrate metabolic activity in the brains of insects[36]. In particular, variation in brain glucose metabolism is apparently linked to variation in aggressive behavior in both *A. mellifera* and *Drosophila melanogaster*[37]. Sucrose metabolism has also been found to play a role in

the regulation of aggressive behavior in the Argentine ant, *Line-pithema humile*[38]. It has been proposed that these fluctuations in aggression in response to metabolic changes could be mediated through variation in mitochondrial energetics in the brain[36]. The role of carbohydrate metabolism in mediating aggression has even been observed in vertebrate systems such as sticklebacks and mouse[8] indicating the possibility of conserved or convergent gene networks underlying aggression. Our differential expression data are consistent with and further support the interplay between metabolism and aggression in insects.

The most significantly enriched GO term among differentially expressed genes was "oxidation–reduction process". Enrichment of this term has been observed among genes with expression differences linked to nourishment level and caste in *Polistes* wasps[39,40]. Genes relating to oxidation–reduction processes have also consistently been implicated in differential aggression in the honey bee *A. mellifera*[35,41]. Our size-matched *C. calcarata* offspring raised with and without a mother both received equivalent mass provisioning at the beginning of development, but nourishment may still be a confounding factor (prior experiments have demonstrated changes in aggression in response to diminished nutrient rations in *C. calcarata*[42]).

Eleven of the differentially expressed genes we identified are affiliated with the ionotropic glutamate receptor (IGR) signaling pathway, roughly three times the number expected by chance. The IGR pathway is involved in synaptic communication generally as well as odor perception in specific IGR subfamilies[43]. This class of genes offers a possible neurological explanation for the change in behavior observed in offspring raised without a mother. Fluctuations in IGR expression could be a sign of neural plasticity[44], manipulating how bees respond to environmental stimuli. IGRs have even been linked to aggression phenotypes specifically in *A. mellifera*[41].

We further leveraged our RNA-sequencing data to search for alternatively spliced transcripts between care and no-care individuals. We found hundreds of genes containing significantly differentially utilized exons, a simple marker of alternative splicing. Among these genes, we observed functional enrichment of oxidation–reduction processes as well as spermine biosynthesis and protein phosphorylation (Supplementary Data 1). These alternatively spliced genes may play important roles in the behavioral plasticity we observed in *C. calcarata*, as a role for alternative splicing in developmental plasticity has been documented in many species[29].

In conjunction with RNA-sequencing analysis, we utilized bisulfite sequencing to quantify changes in DNA methylation in response to the loss of maternal care. We detected localized but significant changes in DNA methylation between the treatment groups, demonstrating that such differences can be detected among behaviorally distinct conspecific hymenopterans. We also confirmed previous findings that genes containing DNA methylation are more likely to be alternatively spliced than genes lacking DNA methylation (Chi-square test, $p = 4.95e-06$; Supplementary Table 7). However, we did not detect a significant association between *differential* methylation and alternative splicing (Supplementary Table 6). This suggests that a dynamic relationship between DNA methylation and splicing may not exist in *C. calcarata*. At minimum, it is clear from our data that changes in splicing corresponding to changes in DNA methylation are the exception rather than the rule. Changes in DNA methylation do not appear to consistently lead to changes in splicing, nor vice versa, at a level observable in our data set.

What explains the lack of global concordance between variation in DNA methylation and variation in transcription in our data? One factor may be that DNA methylation acts through intermediaries to affect transcription. In particular, methylation-sensitive transcription factors have been documented in mammalian systems[23,45,46] and have been shown to influence both gene expression and splicing[22,45–47]. In such instances, transcription factor expression profiles or splicing patterns could have a profound effect on when and in which cell types the effect of DNA methylation at a binding site would be observed. Our study investigates these associations in a restricted subset of tissues at a single developmental time point. Furthermore, many sites of DNA methylation may not bind transcription factors in any context, which would preclude gene regulation by transcription factor-mediated mechanisms at such loci.

When we examined the spatial relationship between DMRs and alternative splicing events, we found that three of eight genes with both DMRs and differentially utilized exons exhibited spatial overlap between the DMRs and alternative exon usage (Fig. 4b–d, Supplementary Figure 2). We found no genes in which a DMR directly overlapped any of the other splicing events we assessed (Supplementary Figure 3). Thus, our integrative approach to transcriptome and methylome analysis revealed three candidates for the potential interaction of DNA methylation and alternative splicing. These particular genes encode proteins with putative roles in fructose metabolism (PFKFB1[48]), apoptosis and stress (SLK[49]), and brain development (PHGDH[50]), each of which stands to influence organismal behavior. One of these genes, *6-phosphofructo-2-kinase* (PFKFB1), was the sole gene to exhibit significant differential expression, alternative splicing, and differential methylation in response to removal of maternal care. A second gene, *sh2b adaptor protein 1-like* (SH2B1[51]), also exhibits significant differential expression, alternative splicing, and differential methylation in response to removal of maternal care but does not exhibit direct overlap between differential methylation and alternative splicing event locations. SH2B1 has a putative role in growth factor signaling. These loci represent prime candidates for follow-up investigations of tissue-specific transcription and methylation over the course of *C. calcarata* development.

In conclusion, removal of the mother during early *C. calcarata* development culminates in offspring that exhibit increased aggression and avoidance behaviors, have over one thousand differentially expressed genes, have hundreds of alternatively spliced genes, and exhibit subtle changes in DNA methylation. We identified specific functional groups of genes that may play a role in developmental plasticity and aggression: carbohydrate metabolism genes, oxidation–reduction process genes, and IGR genes. We observed no genome-wide association between differential DNA methylation and differential gene expression or splicing, though we cannot rule out more complex, indirect relationships between these factors. Our study offers unique insights into the effects of maternal care on offspring development and explores understudied questions about the role of DNA methylation in transcriptional regulation in invertebrate taxa. Further exploration of the candidate genes we identified may help to explain the genetic underpinnings of social behaviors in *C. calcarata* and provide insight into how these genes are regulated in response to environmental stimuli.

## Methods

**Sample collection.** Nests of our study species, *C. calcarata*, were collected from naturally occurring dead broken sumac (*Rhus typhina*) stems in Durham, NH in June 2015 during the active brood rearing early phase of the colony cycle[9,10]. Nests were collected from the field so that mothers could forage and feed offspring as they normally would in the wild. Stems were collected prior to 8 am to ensure the mother would be home. Nest entrances were covered with masking tape and transported to the lab where they were split lengthwise to record all nest contents. During this early active brood phase mothers provision eggs with pollen balls and offspring were no older than small larvae. Pollen masses with eggs were reared in lab with entire nests at room temperature until eclosion of adult offspring in

August 2015. To test the effect of maternal care on developing offspring, we generated "care" and "no care" treatment groups among the lab-reared nests for comparison. In the "care" treatment, mothers were left with the nest and offspring were raised from egg to adulthood in the presence of maternal care. In the "no care" treatment, mothers were removed from the nest after the initial mass provisioning and offspring were raised from egg to adulthood in the absence of maternal care. Pollen and eggs were retained so this was not a manipulation of maternal investment, but rather in half of the lab-reared nests mothers were removed and in the other half mothers remained, such that they could engage in offspring grooming, a stereotypical behavior they complete each evening (reviewed in Rehan et al[10]). In the following experiments, we assayed one age- and size-matched adult female offspring per nest to determine the behavioral effects of early termination of maternal care.

**Behavioral assays.** Circle tubes are a well-established method for quantifying behavior in this and other bee species[11,52]. Interactions among individuals in circle tube arenas (e.g. aggression, avoidance, and tolerance behaviors) parallel those expressed in observation colonies[53].

Two dark-winged (five-day-old) offspring bees from different nests were paint marked for unique identification and introduced simultaneously into opposite ends of a clean plastic circle tube[54]. A new piece of tubing was used for each dyad. Observations of each pair lasted for 20 min. An encounter between bees was recorded when individuals came within one body length of each other[55], and aggression and avoidance behaviors were scored[11]. We conducted 52 circle tube trials. Two females from different nests were paired and no female was assayed twice. Therefore, 104 female offspring in total were assayed in this experiment.

**RNA-sequencing and quantification.** For each sample, RNA was extracted from the heads of three adult female bees and pooled together. Heads were utilized to ensure comparability with bisulfite sequencing data, which have high DNA input demands (see below), and a pooling strategy was leveraged to minimize individual variation in gene expression and maximize our ability to detect the effect of maternal care specifically. All bees utilized for this analysis were from different nests. Each pool was then subject to an Illumina Truseq library preparation protocol to provide a genome-wide profile of mRNA transcript abundance. Three biological replicates were prepared from individuals reared in the presence of the mother and three biological replicates were prepared from individuals reared in the absence of a mother. We note that our statistical power and ability to detect weakly differentially expressed genes is dampened relative to an experimental design featuring more biological replicates, and the sequencing of RNA and DNA from multiple tissues (heads) dampens our data's sensitivity to tissue-specific differences. Library sequencing was performed on an Illumina HiSeq platform by Genome Quebec (Montreal). Paired-end, 100 bp reads were trimmed using Trimmomatic (v0.32)[56]. Reads were then aligned to the *C. calcarata* reference assembly (v1.1, GCA_001652005.1)[13] using Tophat[57] with parameters "–no-mixed–no-discordant" to filter out read pairs that did not have both reads mapping or had a read pair mapping discordantly. Each sample produced between 92 and 109 million aligned reads. A target of one hundred million mapped reads per sample is considered much greater depth than necessary for differential expression analysis, but was chosen for this project to enhance detection of alternative splicing[58]. At this point, multiple pipelines were utilized, each using the Tophat output as input.

**Computing differential expression with edgeR.** Aligned reads from each library were mapped to the existing gene models[13] using Rsubread featureCounts[59]. Resulting counts files were then merged and input into edgeR[60]. Genes with more than three libraries being represented by 0.18 counts per million reads, translating to at least 10 reads aligned to a gene, were kept for further analysis. RPKM values were calculated for the remaining genes and the edgeR Exact Test was used to determine which genes were differentially expressed. Differences were considered significant if they had an FDR-corrected *p*-value <0.01.

**Computing differential exon usage with DEXseq.** Aligned reads from each library were mapped to the existing gene models using the script provided by DEXseq[61]. DEXseq was then used with the standard implementation to compute differential exon usage between the maternal care and no care groups. Exons were considered differentially utilized if they had an FDR-corrected *p*-value <0.01. The native plotting function in DEXseq was used to plot the exon usage of genes that contained a differentially utilized exon as well as a DMR.

**Additional alternative splicing metrics from rMATS.** Aligned reads spanning junctions were quantified using rMATS[62]. Genes containing an alternative splicing event with an FDR-corrected *p*-value <0.01 were considered significantly alternatively spliced. Alternative splicing events in genes that also contained DMRs were visualized using the DEXseq plotting function. rMATS is ideal for identifying a variety of alternative splicing event types, but its dependence on junction-spanning reads makes it liable to miss alternative splicing events that occur in low proportion. DEXseq, while only able to detect specific types of splicing events, leverages more total reads to yield a more thorough analysis of differential exon usage.

**Bisulfite sequencing and quantification.** Genomic DNA was extracted and pooled from the heads of nine adult females from different nests for each sample used for bisulfite conversion and Illumina sequencing, in order to provide a nucleotide-resolution genomic profile of DNA methylation. Pooling was used to minimize the effects of individual variation while heads were used to accommodate the high DNA input requirement for bisulfite conversion. Three biological replicates each were sequenced (90 bp, paired-end reads after barcode removal) from individuals raised in the presence of maternal care and from individuals raised in the absence of maternal care. Unmethylated Enterobacteria phage lambda DNA (GenBank accession: J02459.1) was added to each sample as a control for bisulfite conversion. Bisulfite conversion and sequencing were performed by BGI (Shenzhen, China) on the Illumina HiSeq platform. We trimmed adapters and removed low-quality reads using Trimmomatic (v0.32)[56]. Reads were then aligned to the reference genome using Bismark (v0.14.5)[63]. Bismark was also used to remove duplicate reads and to extract and quantify the per-site methylation level. We determined that any strand-asymmetric methylation (hemimethylation) was likely due to low coverage (Supplementary Figure 1A, B), and with this information in mind, we merged the strands so that each CpG dinucleotide was represented by one methylation value. We generated an average coverage depth between 21X and 37X reads in each sample at each strand merged CpG. The nonconversion rate was then computed based upon the methylation level in the lambda DNA and this value was used to compute the methylation status for each site based upon an FDR-corrected Binomial test performed in R.

A global methylation level for each gene was computed as the total number of methylated (unconverted) reads summed over all CpG sites divided by the total number of reads summed over all CpG sites. These levels were then plotted to determine a reasonable cutoff for a gene to be considered "methylated" (Supplementary Figure 1C). We chose global CpG methylation (all methylated CpG reads/all CpG reads) >5% for a given gene as a reasonable cutoff. All genes passing this cutoff in at least one of our six samples were compared against a background of the full *C. calcarata* gene set using TopGO. Global methylation and CpG density (total number of CpGs/total number of nucleotides) were computed for each gene feature and a metaplot was created (Supplementary Figure 1D).

**Differential methylation analysis.** After extraction of the methylation levels for each CpG, the values were filtered down to only those CpGs that had significant methylation in at least one sample (FDR-corrected binomial test *p* < 0.05 with null expectation set by the nonconversion rate of the bisulfite reaction). This was done to diminish background noise in the differential methylation analysis. Following this filtration, MethylKit[64] was used to compute the differential methylation between the "no maternal care" and "maternal care" samples. CpGs with no coverage in any sample were removed, as were CpGs with coverage >170 in any sample to remove potential PCR artifacts. CpGs were then binned into 200 bp windows and windows that contained fewer than 2 CpGs were removed. The median exon length in *C. calcarata* is 176 bp, while the mean is 355 bp; we used 200 bp windows to approximate exon size so that we could readily map differential methylation to genomic features. Finally, differential methylation was computed for each region using Benjamini–Hochberg FDR correction[65] with a threshold set as minimum methylation difference of 5% and maximum FDR-corrected *p*-value of 0.01. Significant DMRs were then checked for genic localization and a list of genes containing DMRs was created.

In order to ensure that the differential DNA methylation we detected was driven by biological factors, differential methylation analysis was also performed on the nine possible permutations of sample ID assignment, shuffling "maternal care" and "no maternal care" identity using a modified QAM[66]. For this QAM analysis, each 200 bp genomic region was given a probability of being a DMR based on how many of the nine permutations yielded a significant result at that locus. Then 50,000 random, region-independent permutations of the analyses were generated, and the number of DMRs in each iteration was tallied. This method is similar to the permutation method used in Libbrecht et al.[67], except we utilized 200 bp windows rather than individual CpGs and computed differential methylation for each permutation using MethylKit[64] rather than a series of binomial tests on each sample. These modifications allow us to robustly identify DMRs even when coverage is low in some samples at the given region.

**Exploring and comparing the significant gene sets.** The package UpSetR[68] was used to compute the overlap between the lists of significant genes from analyses of differential expression, alternative splicing, and differential methylation. The R package "pvclust"[69] was used to determine the degree of clustering between the two treatment groups based on gene expression and genic methylation. GO term annotations from a previous study[13] were used to assess GO term enrichment for each list of significant genes using the default topGO[70] "weight01" method and the "fisher" statistic. GO terms were considered significantly enriched if they had a *p*-value <0.05. We also mapped *D. melanogaster* orthologs of *C. calcarata* genes using a reciprocal best BLAST hit approach and imported Online Mendelian Inheritance in Man phenotype information associated with these fly orthologs for perusal (Supplementary Data 2).

**Data availability**. Raw reads and analyzed data for both the RNA-sequencing and bisulfite sequencing are available at the Gene Expression Omnibus accession GSE111611. All other data supporting the findings of this study are available within the paper and its supplementary files.

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

## Acknowledgements

This work was supported by the University of New Hampshire and the University of Georgia College of Agricultural and Environmental Sciences. We thank Wyatt Shell for the illustrations used in Fig. 1 and Karl Glastad for bioinformatic consultation.

## Author contributions

S.M.R. conceived of the study. S.M.R. and B.G.H. designed the study. S.M.R. collected specimens, performed rearing experiments, performed behavioral assays, and prepared samples for sequencing. S.V.A., B.G.H., and S.M.R. conceived of bioinformatic analyses; S.V.A. performed them. S.V.A. wrote the manuscript with feedback and contributions from B.G.H. and S.M.R.

## Additional information

**Competing interests:** The authors declare no competing interests.

