## [Peer Review file · Nature Communications]

Reviewers' comments:

Reviewer #1 (Remarks to the Author):

This manuscript uses a well-established behavioral assay to address the question of early experience effects on aggression, transcriptomics and epigenetics in a subsocial bee. There is a strong interest in this topic, so the paper potentially can make an important contribution.

I have the following concerns.

This question has been addressed in mammals, and it is thus not clear why in this species? The authors need to make a case for the importance of such a comparative analysis. The case can be made and Rehan is an accomplished researcher in this area and so should be able to make it.

P3 Were nests moved to the lab and then maintained there before circle tube assays performed? How did this work? More detail needed.

P4 The samples used for the omics analyses are unfortunately not reflective of best practices. Using heads instead of brains diffuses the impact and interpretation of the results. Similarly, pooling individuals introduces more variation than is deal and variation that cannot be modeled statistically. Finally the small Ns are not impressive.

The omics analyses are leveraged off of aggressive interactions. But the study apparently ignored another potent influence on aggression in addition to early experience of queenlessness –body size. Was body size controlled for or measured in staging the circle tube encounters? If not, why not?

Reviewer #2 (Remarks to the Author):

Here Arsenault et al. examine the association of gene expression, alternative exon usage, and DNA methylation levels with alternative adult behavioral phenotypes in a carpenter bee. Major outcomes include (1) maternal care is important for offspring adult phenotypes, (2) gene expression, alternative splicing, and methylation (at a small number of genes, for the latter) is associated with those phenotypes, (3) there's more or less no overlap between differentially expressed genes, differentially splicing genes, and differentially methylated genes (a single candidate between all three).

The manuscript is generally interesting and well-written. The results are largely correlational and identify candidates for future functional work. It will be of most interest to researchers curious about the physiological basis of aggressive versus less aggressive phenotypes. For example, the study reveals that expression differences in the ionotropic glutamate receptor genes may be particularly important. The study does little to advance our understanding of

the link between gene expression and DNA methylation in insects, or between phenotypic plasticity and DNA methylation.

I have a number of questions and comments detailed below. My biggest concern, experimentally, was about genotypic variation in the experiments and how it was distributed across samples. This is potentially a confounding problem and I did not see it addressed anywhere in the manuscript. This is easily fixed with text if the genotypes were distributed randomly.

More detailed questions and comments:

How much work has been done on maternal care and its impact on aggression in offspring in insects or in this species specifically? They seemed to have reason to suspect that different aggression outcomes would occur. Is their behavioral work a novel result?

Is this care or lack of care something that happens commonly to these bees in nature? Is there any reason to believe that their genomes would have a concerted response to this situation? Or is this completely novel to them? Expectations could be different under each scenario.

The manuscript is set in the context of how little is known about the mechanisms of developmental plasticity. This study does not address this problem, given that they are examining the phenotypic outcomes of plasticity, not how the plasticity itself works.

Lines 41-2: This is overstated. There are quite a few studies revealing the mechanisms of developmental plasticity. See *Pristionchus* studies for excellent examples.

Lines 94-107: The sampling is unclear. There were a set of groups of offspring in the same nest without mothers and another set of groups with mothers. How were these offspring distributed into the RNA-Seq and DNA-Seq experiments? Were there cases of multiple individuals from the same nest going into a treatment? My question is ultimately about genotypic effects on these experiments. How much genotypic variation is there amongst individuals/nests/etc and could any systematic bias impact results?

I'm not familiar with DEXseq specifically. How many reads are required to cover a junction in order to call a differential splice site? Why focus on alternative exon usage and not other forms of alternative splicing?

Line 178: What's "significant methylation" here? And what's the rationale for using 200bp windows for the binning?

Not sure I am understanding their differential methylation calling. Is it fundamentally different than the method used in a key paper in the literature, which they cite (63)? If it's different, is it more or less conservative? Calling of differentially methylated DNA seems to be controversial, so methods should be justified and addressed.

Figure 1: They performed a nonparametric MWU test, but then show means in 1C. For

consistency and data transparency, can they please show a box plot in 1C with individual data points also visible?

And again, because it's not clear how many individuals per nest were used in the behavioral experiment, it's not clear whether or not there was any pseudoreplication due to use of siblings/closely related genotypes.

Line 234: It would help to state why they would expect differentially expressed genes to show signs of positive selection. This seems like an extraneous analysis, as is. Same for line 247.

I'm not sure the GO enrichment terms are interesting enough to warrant inclusion in a non-supplemental table.

Figure 3: What does "Methylation Difference" mean on the x-axis in 3B (same thing in figure 4)? Is this in percentages? And, again, I'm having a little trouble understanding the methods behind 3C. Is it correct that about 50 of their observed 203 differentially methylated windows could be spurious? Could they please address this in the text where they refer to 3C?

Is there any reason to think that a change in methylation by 5% or even 20% would be functional? It would help the reader to know what has been shown previously.

Overall, was there more methylation in maternal care offspring?

"These alternatively spliced genes may play important roles in the behavioral plasticity we observed in *C. calcarata*, as a role for alternative splicing in developmental plasticity has been well documented in many species⁶²." This reference is strangely used, since this cited paper is putting forth the argument that alternative splicing is not well studied in phenotypic plasticity, but should be.

Reference 63 is used oddly used as well, since that study showed no differences.

Near the end they reiterate that this treatment has resulted in over one thousand differentially expressed genes. They don't specifically say that this seems remarkable, but it is almost implied. It's becoming more and more clear that large-scale gene expression changes can distinguish individuals. Moreover, they have likely included in their dataset a fair number of very lowly expressed genes (given their high sequence coverage and non-conservative cutoff of 10 reads/gene in three libraries). These would not be picked up in most RNA-Seq differential expression studies, further increasing their numbers.

Reviewer #3 (Remarks to the Author):

This study investigated how maternal care (and the absence of a mother) impacts adult

offspring behavior in the small carpenter bee. In addition to evaluating behavioral effects, the study assessed differences in whole-head transcriptomes and methylomes for individuals kept in nests with and without maternal care.

In general, I think this study is of broad interest, and includes several important analytical approaches to understanding how the developmental environment impacts behavior. I recognize the extensive effort that went into data collection and analysis. That being said, I had difficulty interpreting the biological significance of the study. For one, there were critical experimental design details missing that made it difficult to understand the nature of maternal care for the life stages evaluated, the conditions under which individuals were kept in the lab, and the interpretation of the behavioral tests in the adult bees (see extensive comments below). I also struggled to understand the interpretation of the methylation data, as it is periodically a bit unclear if the authors are discussing broad links between methylation and gene expression (in the genome), or focused on evaluating the source of differentially expressed genes across treatments. The general role of DNA methylation, and the role it plays in explaining environmentally-induced differential gene expression, could be fundamentally different – the distinction is unclear at times. I am not sure how to interpret the significance of the methylation results, where there was little evidence for differences in global methylation patterns, but nonetheless some regions that were differentially methylated – I am wondering if this is ascertainment bias, as the DMRs were random windows defined a priori as showing treatment differences. I am not sure of the gene content and how it varies across windows. One interpretation that is not given much weight is that methylation is not playing a critical role, either in mediating differential gene expression or behavioral expression. Finally, in terms of general significance of the study, I thought the authors could do more to justify the knowledge gaps in the field of developmental plasticity. This is a well-studied phenomenon in a variety of taxa including insects.

Please see specific comments below.

L41-42 – It would be good to qualify the gaps in knowledge. I can think of some systems where there is a fair bit known about developmental plasticity.

L69 – This is a bit confusing as worded (positive effect on transcription, suppressing elongation, etc), possibly because of differences in how these data were collected (e.g., the first seems like it was a genome wide survey of transcripts and methylation patterns, and the second was for de novo transcription somehow measured in real time?). Maybe you can add something in here to clarify the apparent contradiction, or try to reconcile it (or maybe you think this hypothesis is incorrect, which you could also state).

L73 – It affects transcription factor binding in gene bodies, or upstream in regulatory regions? Confusing as written.

L74 – Can you explain "weak splice sites"?

L75 – By mechanistic, do you mean beyond genome-wide correlations, or something else?

Maybe you can specify exactly what you mean by mechanistic.

L79 – By the end of the Introduction it sounds like you are going to test these alternative hypotheses about the action of DNA methylation in insects, but I'm not sure that is primarily what you are going to do.

L99 – I think it should be "eggs with pollen balls"?

L101 – I am having a hard time imagining the lab set up in this experiment. Were the mothers and offspring in some sort of container connected to the outside, or were they unable to leave the lab? Were mothers still provisioning offspring at this point, or was that step completed prior to the start of the experiment? If the latter, how do you know? How many offspring were manipulated per mother, one, or did a mother care for multiple of her own offspring? Were the offspring still enclosed in a nest of some kind, or just sitting out in the open on their pollen balls? Also, at what stage are individuals typically orphaned in nature? It sounded like in the Introduction it was while they were already adults. Here it was egg through adult?

L106 – How was age determined? Adult emergence time (e.g. adult age) or egg-lay-date? If emergence time, it is possible development rates varied, which makes it hard to interpret what similar adult ages is telling us (i.e., are you focused on the period from egg to adult, or from adult on in terms of developmental effects on offspring?)

L117 – 52 assays of care versus no care, correct? Out of how many, and how were individuals selected? (e.g., did you assay everyone with their nearest size and age matched individual, or were some left out?) It would be good to have sample sizes up in the previous paragraph as well.

L118 – what is the logic of matching non-nestmates in the aggression assay? It seems like the normal context for aggression is against siblings (based on the Introduction)? How was relatedness among contestants accounted for? Can you provide more details about the lab conditions to assess potential sources of differential aggression (e.g., are they picking up unique odors?)

L123 – How did you select individuals for RNAseq, given the large number of available choices? This is three pooled heads over three replicates, so 9 individuals included total for N=3 per treatment?

L128 – sounds like this was paired end sequencing? How long were the reads?

L131 – I agree you have substantial read numbers here, but depth and replication are both important for differential expression analysis, and a priori, your number of replicates is quite low.

L138-139 – I am a bit confused about the read counting strategy – it seems like you are mapping counts (e.g., you filtered based on counts per million) but then you convert that

into RPKM. Doesn't edgeR usually operate on read count data?

L151 – can you justify why the bisulfite seq had a different pool size (9 heads) than RNAseq (3 heads). I'm assuming tissue quantity, but larger pools may also show lower variance across treatments, so it seems important to justify if you are going to compare RNAseq results to bisulfite seq in any way.

L159 – you may want to list genome version and the source if such a thing exists (I realize there may only be one version).

L161 – can you explain "asymmetric" methylation and its implications? Does that mean among reads in a single sample, across samples in a treatment, across treatments?

L163 – the strand merging piece could possibly be explained a bit more clearly (you may have surmised that I'm not a methylation expert!)

L164 – why the range of coverage? How was that distributed across samples/treatments?

L171 – is this a typical cut-off level?

L178- I'm assuming a CpG is not associated with a gene at a 1:1 level at this point because there may be multiple methylation sites per gene? Can you give an idea of the relationship between number of CpGs and number of genes at this stage (and average number of CpGs per gene)?

L181 – given the threshold cut-off you used for a methylated gene, how are there CpGs that show no coverage in any sample? I'm assuming it is because a gene could be methylated but a particular site is not methylated in any sample? So, going back to the threshold, what is the 5%? 5% of potential CpG sites for a given gene are methylated (so its weighted by the number of CpG sites?), or is the 5% based on some other metric of degree of methylation?

L197 – "each method" is the methylation analysis and the differential expression analysis?

Fig 1 – it looks like intact nests were kept in the lab? I was confused in the methods because you opened the nests to look at contents (but I'm assuming closed them back up)? How did you control for things like birth order, and number of offspring in a nest, in selecting individuals for your various analyses? I'm still unclear on how long adults were alone without a mother prior to testing? Was this the same across nests? What is the mother actually doing during this period of offspring life, especially under lab conditions? Is she actively foraging and bringing food? How did this correspond to age or age differences among offspring? What is frequency of the Y-axis of Fig 1C? Are those counts of interactions, or proportions of interactions? Frequency seems to imply some sort of proportion.

L267 – I now see the answer about the 5% threshold – I guess this could be clearer in the

methods.

L270, L291-292 – Importantly in these assessments you show that differential methylation is not associated with positive selection, but that is different from an explicit test for purifying selection (which are not the only two alternatives). Is there a reason you did not assess evidence of purifying selection directly?

The lack of biological context for the manipulation in the methods makes it hard for me to interpret the strong differences in gene expression and methylation patterns you report. On its face, this degree of effect (especially for differential gene expression) seems surprising – that doesn't mean it is not real, it just requires more context to interpret.

L313 – I feel like it is important here to distinguish whether you are talking about differences across genes in the genome, or differences observed across your experimental treatments. Methylation does seem to explain gene expression variation among genes in the genome, but just not in the context of genes differentially expressed across your treatments. Furthermore, adding the DMR piece gets a little tricky since these were specifically selected as regions that differed across treatments (somewhat random(?) 200 bp regions of the genome). Should the expectation be that genes (or portions of genes) in these regions are functionally related, or expressed as cassettes? How does the degree of methylation correspond to the number of genes in these regions (what's the average methylation per gene in these regions?). Any variation in these values seems like it could obscure the relationship between the DMR and gene expression values. In a similar vein, you have not compared the existence of methylated regions (equivalent to DMR, but randomly selected and not treatment-specific) with gene expression patterns in a global sense, so it is hard to compare the genome wide patterns to treatment level phenomena by incorporating both DMR and global methylation information.

L321 – How to interpret this extremely limited set of genes? Is this just random, or is this a statistically significant relationship between methylation and alternative exon expression?

Reviewers' comments:

Reviewer #1 (Remarks to the Author):

This manuscript uses a well-established behavioral assay to address the question of early experience effects on aggression, transcriptomics and epigenetics in a subsocial bee. There is a strong interest in this topic, so the paper potentially can make an important contribution.

I have the following concerns.

This question has been addressed in mammals, and it is thus not clear why in this species? The authors need to make a case for the importance of such a comparative analysis. The case can be made and Rehan is an accomplished researcher in this area and so should be able to make it.

We have added the following sentence to directly address this point: "However, the gene regulatory consequences of DNA methylation are more poorly understood in insects than in mammals, and it is presently unknown whether DNA methylation mediates potential long term effects of variation in parental care in *C. calcarata* or other subsocial insects."

P3 Were nests moved to the lab and then maintained there before circle tube assays performed? How did this work? More detail needed.

We have added more detail to better explain our experimental design. As stated before, "In the 'care' treatment, mothers were left with the nest and offspring were raised from egg to adulthood in the presence of maternal care. In the 'no care' treatment, mothers were removed from the nest after the initial mass provisioning and offspring were raised from egg to adulthood in the absence of maternal care." We also now state that, "Pollen and eggs were retained so this was not a manipulation of maternal investment but rather in half of the lab reared nests mothers were removed and in the other half of nests mothers remained such that they could engage in offspring grooming, a stereotypical behavior they complete each evening (reviewed in Rehan et al. 2014)."

Rehan, S. M., Berens, A. J. and Toth, A. L. (2014). At the brink of eusociality: transcriptomic correlates of worker behavior in a small carpenter bee. *BMC Evol. Biol.* 14, 260).

P4 The samples used for the omics analyses are unfortunately not reflective of best practices. Using heads instead of brains diffuses the impact and interpretation of the results. Similarly, pooling individuals introduces more variation than is deal and variation that cannot be modeled statistically. Finally the small Ns are not impressive.

We chose to use whole heads and pooled individuals due to the intensive DNA demands of bisulfite conversion (for example Lyko et al. *PLOS Biol* 2010 pooled 50 honey bee brains per sample). In order to generate comparable RNA-seq data, the same approach was taken here. We agree that this approach has drawbacks, but we still believe our study yields insights of general interest, and there remains a scarcity of published DNA methylation studies in insects with biological replication at the level of $n=3$.

The omics analyses are leveraged off of aggressive interactions. But the study apparently ignored another potent influence on aggression in addition to early experience of queenlessness –body size. Was body size controlled for or

measured in staging the circle tube encounters? If not, why not?

As noted in text, “we assayed age and size-matched adult female offspring to determine the behavioral effects of early termination of maternal care.” (from Methods, *Sample collection*)

Reviewer #2 (Remarks to the Author):

Here Arsenault et al. examine the association of gene expression, alternative exon usage, and DNA methylation levels with alternative adult behavioral phenotypes in a carpenter bee. Major outcomes include (1) maternal care is important for offspring adult phenotypes, (2) gene expression, alternative splicing, and methylation (at a small number of genes, for the latter) is associated with those phenotypes, (3) there’s more or less no overlap between differentially expressed genes, differentially splicing genes, and differentially methylated genes (a single candidate between all three).

The manuscript is generally interesting and well-written. The results are largely correlational and identify candidates for future functional work. It will be of most interest to researchers curious about the physiological basis of aggressive versus less aggressive phenotypes. For example, the study reveals that expression differences in the ionotropic glutamate receptor genes may be particularly important. The study does little to advance our understanding of the link between gene expression and DNA methylation in insects, or between phenotypic plasticity and DNA methylation.

We respectfully disagree with the reviewer’s statement that “the study does little to advance our understanding of the link between gene expression and DNA methylation in insects.” We believe that we present an important result – that in replicated samples with significant gene expression and DNA methylation differences, there is little observable relationship between differential DNA methylation and differential gene expression or splicing. Although this lack of overall association does not directly implicate DNA methylation in transcriptional regulation, the lack of concordance between variation in DNA methylation and variation in transcription is an important finding that directly helps to “advance our understanding of the link between gene expression and DNA methylation in insects” by showing that DNA methylation plays at most a minor role in overall transcriptional regulation in *C. calcarata*.

I have a number of questions and comments detailed below. My biggest concern, experimentally, was about genotypic variation in the experiments and how it was distributed across samples. This is potentially a confounding problem and I did not see it addressed anywhere in the manuscript. This is easily fixed with text if the genotypes were distributed randomly.

In all of our assays (behavioral and molecular) we used bees from different nests in order to control for genotypic variation as much as possible. This drastically limits the possibility that we assayed siblings in our behavioral assays or have confounding factors due to genotype in our molecular assays. Additionally, in both the RNA-seq and bisulfite-sequencing assays, we used multiple pooled individuals from different nests for each replicate which further dilutes the possibility of genotypic variation contributing to the effects we are observing. We have modified the methods text to better reflect this facet of our experimental design.

More detailed questions and comments:

How much work has been done on maternal care and its impact on aggression in offspring in insects or in this species specifically? They seemed to have reason to suspect that different aggression outcomes would occur. Is their behavioral work a novel result?

Maternal interaction is relatively rare in insects. Among subsocial and social insects, maternal aggression is relatively common. In eusocial species mothers typically coerce offspring (workers) into foraging and repeated physical coercion is known to reduce ovarian development (Brothers and Michener 1974; Buckle 1982). In *Ceratina calcarata* specifically there is evidence of maternal coercion of offspring (Rehan and Richards 2013). Thus, aggression is well established means of establishing dominance hierarchies in this and other species (reviewed in Withee and Rehan 2016).

Brothers, D. J. and Michener, C. D. (1974). Interactions in colonies of primitively social bees- III. Ethometry of division of labour in *Lasioglossum zephyrum* (Hymenoptera: Halictidae). *J. Comp. Physiol.* 90, 129-168.

Buckle, G. R. (1982). Differentiation of queens and nestmate interactions in newly established colonies of *Lasioglossum zephyrum* (Hymenoptera: Halictidae). *Sociobiology* 7, 8-20.

Rehan, S. M. and Richards, M. H. (2013). Reproductive aggression and nestmate recognition in a subsocial bee. *Anim. Behav.* 85, 733-741.

Withee, J. R. and Rehan, S. M. (2016). Cumulative effects of body size and social experience on aggressive behaviour in a subsocial bee. *Behav.* 153, 1365-1385.

Is this care or lack of care something that happens commonly to these bees in nature? Is there any reason to believe that their genomes would have a concerted response to this situation? Or is this completely novel to them? Expectations could be different under each scenario.

This bee normally provides prolonged maternal care. The absence of care is the ancestral state of bees since most bees are solitary. We had no prior expectation on the genomic response to maternal care or lack thereof but interestingly found a striking response.

The manuscript is set in the context of how little is known about the mechanisms of developmental plasticity. This study does not address this problem, given that they are examining the phenotypic outcomes of plasticity, not how the plasticity itself works.

Lines 41-2: This is overstated. There are quite a few studies revealing the mechanisms of developmental plasticity. See *Pristionchus* studies for excellent examples.

The reviewer makes an excellent point, and we have removed language suggesting that little is known about developmental plasticity in a slight reframing of the introduction.

Lines 94-107: The sampling is unclear. There were a set of groups of offspring in the same nest without mothers and another set of groups with mothers. How were these offspring distributed into the RNA-Seq and DNA-Seq experiments? Were there cases of multiple individuals from the same nest going into a treatment? My question is ultimately about genotypic effects on these experiments. How much genotypic variation is there amongst individuals/nests/etc and could any systematic bias impact results?

For all of the behavioral and molecular assays, we used individuals from different nests to control for (and randomly distribute) genotypic variation. We have clarified this point in the methods section.

I'm not familiar with DEXseq specifically. How many reads are required to cover a junction in order to call a differential splice site? Why focus on alternative exon usage and not other forms of alternative splicing?

The differential exon utilization analysis in DEXseq utilizes differential read coverage among exons more than junction reads. However, we have now performed an additional analysis of various types of alternative splicing events with the tool rMATS, which utilizes junction reads. We have incorporated these new results into Figure 4 and supplementary material.

Line 178: What's "significant methylation" here? And what's the rationale for using 200bp windows for the binning?

We have now clarified in the methods text that significant methylation refers to "FDR-corrected binomial test $P < 0.05$ with null expectation set by the nonconversion rate of the bisulfite reaction". We used a windowed approach as opposed to single CpGs because extremely high coverage is necessary to find significant differentially methylated CpGs after FDR-correction due to the drastically increased number of tests that have to be performed to quantify differential methylation analyses at single CpG resolution. That being said, we wanted to get a relatively fine scale view of the methylation landscape so we used 200bp windows, which almost always contain more than one informative CpG while also approximate exon size so that we can determine which genomic feature exhibits differential methylation. For instance, the median exon in *C. calcarata* is 176bp long (mean is 355bp). Additionally, a single nucleosome generally spans approximately 146bp of DNA so a 200bp window approximates a single nucleosome or average sized exon while keeping the number of differential methylation tests we had to perform to a more reasonable number given our coverage. We have added text to the methods to clarify.

Not sure I am understanding their differential methylation calling. Is it fundamentally different than the method used in a key paper in the literature, which they cite (63)? If it's different, is it more or less conservative? Calling of differentially methylated DNA seems to be controversial, so methods should be justified and addressed.

We have added the following text to clarify: "This method is similar to the permutation method used in Libbrecht et al. except we utilized 200bp windows rather than individual CpGs and computed differential methylation for each permutation using MethylKit rather than a series of binomial tests on each sample. These modifications allow us to robustly identify differentially methylated regions even when coverage is low in some samples at the given region."

Figure 1: They performed a nonparametric MWU test, but then show means in 1C. For consistency and data transparency, can they please show a box plot in 1C with individual data points also visible?

We have replaced the barplots in panel 1C with box plots and added additional information about box plot ranges and sample sizes to the legend.

And again, because it's not clear how many individuals per nest were used in the behavioral experiment, it's not clear whether or not there was any pseudoreplication due to use of siblings/closely related genotypes.

As noted above, each bee came from a different nest.

Line 234: It would help to state why they would expect differentially expressed genes to show signs of positive selection. This seems like an extraneous analysis, as is. Same for line 247.

We have added the following sentence to clarify the justification for this analysis, which applies to both instances: "Given the importance of maternal care in *C. calcarata*, we hypothesized that genes developmentally regulated by maternal care may also be subject to positive selection within the *Ceratina calcarata* lineage."

I'm not sure the GO enrichment terms are interesting enough to warrant inclusion in a non-supplemental table.

We appreciate the suggestion, but we think the GO enrichment table helps to provide a more complete view of our study if one were to read only the abstract and go through the figures and tables. We prefer to keep this as a main text table.

Figure 3: What does "Methylation Difference" mean on the x-axis in 3B (same thing in figure 4)? Is this in percentages? And, again, I'm having a little trouble understanding the methods behind 3C. Is it correct that about 50 of their observed 203 differentially methylated windows could be spurious? Could they please address this in the text where they refer to 3C?

We have clarified in the figure legend that the x-axis in 3B is a percentage. We have also clarified in text that 3C allows us to estimate how likely any of our individual DMRs occurred by chance (18.5%).

Is there any reason to think that a change in methylation by 5% or even 20% would be functional? It would help the reader to know what has been shown previously.

This is an excellent question. Given the frequently low DNA methylation levels in holometabolous insects, it is possible that any change in methylation could be functional (or not). There is very little functional evidence for the role of methylation in these systems. We use a 5% change to quantify differential methylation as it showed the best ability to discriminate DMR's in the given treatment as opposed to permuted versions. Lower values allow too much stochasticity due to coverage and non-conversion while higher values filter out many possible DMRs without addressing the possible problem of coverage differences.

Overall, was there more methylation in maternal care offspring?

Among differentially methylated regions, most (84%) were more highly methylated in individuals reared with parental care. However, these localized changes didn't translate into measurably higher methylation genome-wide in maternally cared-for individuals.

"These alternatively spliced genes may play important roles in the behavioral plasticity we observed in *C. calcarata*, as a role for alternative splicing in developmental plasticity has been well documented in many species⁶²." This reference is strangely used, since this cited paper is putting forth the argument that alternative splicing is not well studied in phenotypic plasticity, but should be.

We have replaced this reference with a more appropriate reference: Kalsotra, A. & Cooper, T. A. Functional consequences of developmentally regulated alternative splicing. *Nat. Rev. Genet.* **12**, 715–729 (2011).

Reference 63 is used oddly used as well, since that study showed no differences.

We have removed this reference for this point as suggested. It is cited instead in the methods to support discussion of differential methylation methodology.

Near the end they reiterate that this treatment has resulted in over one thousand differentially expressed genes. They don't specifically say that this seems remarkable, but it is almost implied. It's becoming more and more clear that large-scale gene expression changes can distinguish individuals. Moreover, they have likely included in their dataset a fair number of very lowly expressed genes (given their high sequence coverage and non-conservative cutoff of 10 reads/gene in three libraries). These would not be picked up in most RNA-Seq differential expression studies, further increasing their numbers.

We agree that the numbers of detected differentially expressed genes vary by a number of factors, including sequencing depth, replicate number, and assayed tissue. We do not advocate for comparing differentially expressed genes between studies without controlling for such factors.

Reviewer #3 (Remarks to the Author):

This study investigated how maternal care (and the absence of a mother) impacts adult offspring behavior in the small carpenter bee. In addition to evaluating behavioral effects, the study assessed differences in whole-head transcriptomes and methylomes for individuals kept in nests with and without maternal care.

In general, I think this study is of broad interest, and includes several important analytical approaches to understanding how the developmental environment impacts behavior. I recognize the extensive effort that went into data collection and analysis. That being said, I had difficulty interpreting the biological significance of the study. For one, there were critical experimental design details missing that made it difficult to understand the nature of maternal care for the life stages evaluated, the conditions under which individuals were kept in the lab, and the interpretation of the behavioral tests in the adult bees (see extensive comments below).

We have added details to better describe our experimental design, as detailed in responses to the comments made on this issue below.

I also struggled to understand the interpretation of the methylation data, as it is periodically a bit unclear if the authors are discussing broad links between methylation and gene expression (in the genome), or focused on evaluating the source of differentially expressed genes across treatments. The general role of DNA methylation, and the role it plays in explaining environmentally-induced differential gene expression, could be fundamentally different – the distinction is unclear at times. I am not sure how to interpret the significance of the methylation results, where there was little evidence for differences in global methylation patterns, but nonetheless some regions that were differentially methylated – I am wondering if this is ascertainment bias, as the DMRs were random windows defined a priori as showing treatment differences. I am not sure of the gene content and how it varies across windows. One interpretation that is not given much weight is that methylation is not playing a critical role, either in mediating differential gene expression or behavioral expression.

This study was specifically tailored to investigate the prevalence of *differential* DNA methylation and the relationship between variation in DNA methylation and variation in transcription. We agree that the general transcriptional properties of targets of DNA methylation are of great interest, and we have published a series of studies dedicated to understanding the overall gene regulatory correlates of targets of DNA methylation in hymenopteran insects, irrespective of differential methylation (e.g., Hunt et al. 2013, Glastad et al. 2015, Rehan et al 2016, Glastad et al 2017).

Glastad KM et al. 2017. Variation in DNA Methylation Is Not Consistently Reflected by Sociality in Hymenoptera. *Genome Biology and Evolution*. 9:1687–1698.

Rehan SM, Glastad KM, Lawson SP, Hunt BG. 2016. The genome and methylome of a subsocial small carpenter bee, *Ceratina calcarata*. *Genome Biology and Evolution*. 8:1401–1410.

Glastad KM, Hunt BG, Goodisman MAD. 2015. DNA methylation and chromatin organization in insects: insights from the ant *Camponotus floridanus*. *Genome Biology and Evolution*. 7:931–942.

Hunt BG, Glastad KM, Yi SV, Goodisman MAD. 2013. Patterning and regulatory associations of DNA methylation are mirrored by histone modifications in insects. *Genome Biology and Evolution*. 5:591–598.

We have revised the discussion to give more weight to the interpretation that DNA methylation may not be playing a critical role, as follows. “This suggests that a dynamic relationship between DNA methylation and splicing may not

exist in *C. calcarata*. At minimum, it is clear from our data that changes in splicing corresponding to changes in DNA methylation are the exception rather than the rule. Changes in DNA methylation do not appear to consistently lead to changes in splicing, nor *vice versa*, at a level observable in our data set.”

Finally, in terms of general significance of the study, I thought the authors could do more to justify the knowledge gaps in the field of developmental plasticity. This is a well-studied phenomenon in a variety of taxa including insects.

We have slightly reframed the introduction to more directly focus on effects of variation in parental care, one important vector mediating developmental plasticity.

Please see specific comments below.

L41-42 – It would be good to qualify the gaps in knowledge. I can think of some systems where there is a fair bit known about developmental plasticity.

We have removed the sentence stating that little is known about developmental plasticity and, through revisions made to the introduction, instead focus on specific gaps in knowledge that we address in this study.

L69 – This is a bit confusing as worded (positive effect on transcription, suppressing elongation, etc), possibly because of differences in how these data were collected (e.g., the first seems like it was a genome wide survey of transcripts and methylation patterns, and the second was for de novo transcription somehow measured in real time?). Maybe you can add something in here to clarify the apparent contradiction, or try to reconcile it (or maybe you think this hypothesis is incorrect, which you could also state).

We have revised this introduction to gene body DNA methylation to more clearly characterize this apparent contradiction, as follows. “Despite its supposed suppressive role in the promoter region of genes, DNA methylation within the gene body is, overall, positively correlated with transcription in mammals as well as insects. This positive correlation could be explained by preferential targeting of DNA methylation to actively transcribed genes rather than a direct effect of DNA methylation on transcription. Indeed, the presence of gene body DNA methylation may actually reduce gene expression levels by impeding transcriptional elongation efficiency. Thus, the relationship between gene body DNA methylation and the regulation of gene expression levels remains poorly understood.”

L73 – It affects transcription factor binding in gene bodies, or upstream in regulatory regions? Confusing as written.

The sentence has been updated to clarify, as follows. “It is thought that DNA methylation affects transcription factor binding, which in gene bodies affects RNA Polymerase II processivity and the recognition of variably utilized splice sites.”

L74 – Can you explain "weak splice sites"?

We changed this term to ‘variably utilized splice sites’ to clarify.

L75 – By mechanistic, do you mean beyond genome-wide correlations, or something else? Maybe you can specify exactly what you mean by mechanistic.

We changed this phrase from ‘from a mechanistic standpoint’ to ‘beyond genome-wide correlations’ to help clarify our point in this sentence.

L79 – By the end of the Introduction it sounds like you are going to test these alternative hypotheses about the action of DNA methylation in insects, but I'm not sure that is primarily what you are going to do.

We test whether we are able to detect associations between differential DNA methylation and 1) differential gene expression and 2) differential splicing in our data, as hypothesized.

L99 – I think it should be "eggs with pollen balls"?

Updated as suggested.

L101 – I am having a hard time imagining the lab set up in this experiment. Were the mothers and offspring in some sort of container connected to the outside, or were they unable to leave the lab? Were mothers still provisioning offspring at this point, or was that step completed prior to the start of the experiment? If the latter, how do you know? How many offspring were manipulated per mother, one, or did a mother care for multiple of her own offspring? Were the offspring still enclosed in a nest of some kind, or just sitting out in the open on their pollen balls? Also, at what stage are individuals typically orphaned in nature? It sounded like in the Introduction it was while they were already adults. Here it was egg through adult?

We have added substantial detail to our Methods section entitled *Sample collection*, from the following: Nests were collected from the field such that mothers could forage and feed offspring as they normally would in the wild. Pollen and eggs were retained so this was not a manipulation of maternal investment but rather in half of the lab reared nests mothers were removed and in the other half of nests mothers remained such that they could engage in offspring grooming, a stereotypical behavior they complete each evening (reviewed in Rehan et al. 2014). Entire nests were maintained in the lab and one focal female used for circle tube assays. Mothers are long lived and nest loyal in this species and maternal mortality and nest orphaning is observed but towards the end of the season after adult offspring eclose from the natal nest (Mikát et al. 2017).

Rehan, S. M., Berens, A. J. and Toth, A. L. (2014). At the brink of eusociality: transcriptomic correlates of worker behavior in a small carpenter bee. *BMC Evol. Biol.* 14, 260.

Mikát, M., Franchino, C. and Rehan, S. M. (2017). Sociodemographic variation in foraging behavior and the adaptive significance of worker production in the facultatively social small carpenter bee, *Ceratina calcarata*. *Behav. Ecol. Sociobiol.* 71, 135.

L106 – How was age determined? Adult emergence time (e.g. adult age) or egg-lay-date? If emergence time, it is possible development rates varied, which makes it hard to interpret what similar adult ages is telling us (i.e., are you focused on the period from egg to adult, or from adult on in terms of developmental effects on offspring?)

We age-matched adult offspring by adult emergence date. Here we focused on the recently-eclosed adults to determine possible developmental effects that last into adulthood.

L117 – 52 assays of care versus no care, correct? Out of how many, and how were individuals selected? (e.g., did you assay everyone with their nearest size and age matched individual, or were some left out?) It would be good to have sample sizes up in the previous paragraph as well.

As noted in the main text we assayed 104 females. 52 care and 52 no care. This would equate to 104 nests. Many individuals were not used including males and females that eclosed without an age matched female from a control or treatment nest.

L118 – what is the logic of matching non-nestmates in the aggression assay? It seems like the normal context for aggression is against siblings (based on the Introduction)? How was relatedness among contestants accounted for? Can you provide more details about the lab conditions to assess potential sources of differential aggression (e.g., are they picking up unique odors?)

Each female came from a different nest to avoid genetic effects, shared social environment, or chemical cues.

L123 – How did you select individuals for RNAseq, given the large number of available choices? This is three pooled heads over three replicates, so 9 individuals included total for N=3 per treatment?

That is correct. 9 females were used per treatment for RNA-seq. Females were chosen at random and three pooled per sample.

L128 – sounds like this was paired end sequencing? How long were the reads?

We have added the nature of the reads we generated for both the RNA-seq and BS-seq experiments to their respective sections of the methods. Paired-end, 100bp reads.

L131 – I agree you have substantial read numbers here, but depth and replication are both important for differential expression analysis, and a priori, your number of replicates is quite low.

It would be ideal to have a higher number of biological replicates, and we hope to implement greater replication in future studies, but we believe we have enough statistical power to support the inferences made in our manuscript with the replication present in our study, and that these inferences are of interest to a broad scientific audience.

L138-139 – I am a bit confused about the read counting strategy – it seems like you are mapping counts (e.g., you filtered based on counts per million) but then you convert that into RPKM. Doesn't edgeR usually operate on read count data?

Our implementation of edgeR for differential expression analysis was using counts. RPKM is only used for the clustering analysis (done using pvclust), as the normalization of the RPKM computation helps to minimize the feature-length and library-size effects in the clustering analysis (as per Conesa et al. 2016). We computed RPKM from our counts data using the edgeR native rpkm() function.

L151 – can you justify why the bisulfite seq had a different pool size (9 heads) than RNAseq (3 heads). I'm assuming tissue quantity, but larger pools may also show lower variance across treatments, so it seems important to justify if you are going to compare RNAseq results to bisulfite seq in any way.

Yes, DNA quantity was insufficient with n=3 individuals for standard bisulfite sequencing so we pooled samples to overcome this issue.

L159 – you may want to list genome version and the source if such a thing exists (I realize there may only be one version).

We now include the Genbank assembly accession in addition to the original genome paper reference.

L161 – can you explain "asymmetric" methylation and its implications? Does that mean among reads in a single sample, across samples in a treatment, across treatments?

We now clarify by calling this "strand-asymmetric methylation (hemimethylation)." This refers to DNA methylation that occurs on the CpG in one strand but not on the CpG in the reverse complement strand of DNA.

L163 – the strand merging piece could possibly be explained a bit more clearly (you may have surmised that I'm not a methylation expert!)

See clarification from comment above.

L164 – why the range of coverage? How was that distributed across samples/treatments?

In the previous version of the manuscript, we included a range of coverages based on the CpGs we used for differential methylation analysis that had undergone the filtration described in the first sentence of the "Differential methylation analysis" section. The numbers that make up this range were

C4	C5	C6	N4	N5	N6
33.0425	52.9468	34.6887	43.4477	39.9447	38.7051

Given the position of the sentence in the text, we decided to change to the pre-filtration coverage numbers as these make more sense at the point they are included in the manuscript. These numbers are smaller (as many of the low coverage CpGs are later filtered out) and are described here.

C4	C5	C6	N4	N5	N6
21.6829	38.1852	21.3705	28.5222	24.9617	24.3657

In both, cases, the numbers vary across both treatments with neither treatment showing an obvious difference to the other. Additionally, the coverages do not match up with the clustering pattern we see in Figure 3A which indicates that coverage differences are not driving the clustering.

L171 – is this a typical cut-off level?

As stated in text, "these levels were then plotted to determine a reasonable cutoff for a gene to be considered "methylated" (Figure S1:C). We chose global CpG methylation > 5% for a given gene as a reasonable cutoff." We are

hesitant to say that a single cut-off is typical for the field, but we view this as conservative given the data distributions (see Fig. S1) and given that this is the methylation level combined for all CpGs of a given genomic feature.

L178- I'm assuming a CpG is not associated with a gene at a 1:1 level at this point because there may be multiple methylation sites per gene? Can you give an idea of the relationship between number of CpGs and number of genes at this stage (and average number of CpGs per gene?)?

We limited our analyses to 200bp windows that had 2 CpG sites that passed our filtering criteria (those that were methylated according to a binomial test in at least one sample). Those 200bp windows analyzed have a mean of 4 retained CpGs. One in 16 nucleotides would be expected to be a cytosine in a CpG dinucleotide by chance given a normal distribution.

L181 – given the threshold cut-off you used for a methylated gene, how are there CpGs that show no coverage in any sample? I'm assuming it is because a gene could be methylated but a particular site is not methylated in any sample? So, going back to the threshold, what is the 5%? 5% of potential CpG sites for a given gene are methylated (so its weighted by the number of CpG sites?), or is the 5% based on some other metric of degree of methylation?

The reviewer is correct that, “a gene could be methylated but a particular site is not methylated in any sample.” We have clarified that the cutoff for a given gene is 5% global methylation, which we now clearly define in text as “all methylated CpG reads / all CpG reads” as per Schultz et al. 2012.

Schultz, M. D., Schmitz, R. J. & Ecker, J. R. ‘Leveling’ the playing field for analyses of single-base resolution DNA methylomes. *Trends in Genetics* **28**, 583–585 (2012).

L197 – “each method” is the methylation analysis and the differential expression analysis?

We have clarified that this refers to overlap, “from analyses of differential expression, alternative splicing, and differential methylation.”

Fig 1 – it looks like intact nests were kept in the lab? I was confused in the methods because you opened the nests to look at contents (but I'm assuming closed them back up)? How did you control for things like birth order, and number of offspring in a nest, in selecting individuals for your various analyses? I'm still unclear on how long adults were alone without a mother prior to testing? Was this the same across nests? What is the mother actually doing during this period of offspring life, especially under lab conditions? Is she actively foraging and bringing food? How did this correspond to age or age differences among offspring? What is frequency of the Y-axis of Fig 1C? Are those counts of interactions, or proportions of interactions? Frequency seems to imply some sort of proportion.

As noted above, entire nests were maintained in the lab. Offspring were alone from egg to adulthood in the no care treatment. This was the same across nests. In the care treatment, mothers did not forage but rather guard the nest by day and groom offspring in the evening. One female offspring was selected at random from each nest based on date of eclosion. We have changed the y-axis label to ‘counts; to clarify Fig 1C.

L267 – I now see the answer about the 5% threshold – I guess this could be clearer in the methods.

We have clarified in the methods as described above.

L270, L291-292 – Importantly in these assessments you show that differential methylation is not associated with positive selection, but that is different from an explicit test for purifying selection (which are not the only two alternatives). Is there a reason you did not assess evidence of purifying selection directly?

We did not have population genetic information with sequence polymorphism data, which to our knowledge is needed to perform most robust tests of enhanced purifying selection (such as McDonald-Kreitman type tests).

The lack of biological context for the manipulation in the methods makes it hard for me to interpret the strong differences in gene expression and methylation patterns you report. On its face, this degree of effect (especially for differential gene expression) seems surprising – that doesn't mean it is not real, it just requires more context to interpret.

We have provided additional biological context for the manipulations, as described above.

L313 – I feel like it is important here to distinguish whether you are talking about differences across genes in the genome, or differences observed across your experimental treatments. Methylation does seem to explain gene expression variation among genes in the genome, but just not in the context of genes differentially expressed across your treatments. Furthermore, adding the DMR piece gets a little tricky since these were specifically selected as regions that differed across treatments (somewhat random(?) 200 bp regions of the genome). Should the expectation be that genes (or portions of genes) in these regions are functionally related, or expressed as cassettes? How does the degree of methylation correspond to the number of genes in these regions (what's the average methylation per gene in these regions?). Any variation in these values seems like it could obscure the relationship between the DMR and gene expression values. In a similar vein, you have not compared the existence of methylated regions (equivalent to DMR, but randomly selected and not treatment-specific) with gene expression patterns in a global sense, so it is hard to compare the genome wide patterns to treatment level phenomena by incorporating both DMR and global methylation information.

This study was specifically tailored to investigate the relationship between variation in DNA methylation and variation in transcription across *sample types*. As discussed in a response above, we have already published a series of papers investigating variation in DNA methylation and transcription across different genes of a given insect genome (e.g., Hunt et al. 2013, Glastad et al. 2015, Glastad et al. 2017). We have recapitulated some of those results from data generated for this project in existing Figure S1 and in newly a generated Table S8 expanding upon these analyses, as described in the results and quoted below:

“In each of our samples, we observed similar patterns to those found in the previously generated *C. calcarata* methylome and methylomes from many other hymenopteran taxa (Figure S1, Table S8). In particular, all of the newly generated *C. calcarata* methylomes show a bias towards exons at the 5' end of genes (Figure S1). Methylated genes (>5% mCG/CG) showed GO biological process functional enrichment for ribosome biogenesis, translation, and protein folding (Supplementary Data 1) and were less likely to be under positive selection when compared against unmethylated genes (Chi-square test, $p = 2.43e-07$; Table S3). This corroborates previous results that DNA methylation is biased towards constitutively expressed genes, which tend to be subject to strong purifying selection.”

We chose 200 base windows rather than full genes for differential DNA methylation analysis for two main reasons: 1) *C. calcarata* DNA methylation is not randomly distributed throughout genes but is instead heavily biased to exons in 5' gene regions (Rehan et al. 2016), and 2) 200bp windows approximate exon size so that we can determine which genomic feature exhibits differential methylation (median exon size 176bp in *C. calcarata*; this information has been added to methods).

Glastad KM et al. 2017. Variation in DNA Methylation Is Not Consistently Reflected by Sociality in Hymenoptera. *Genome Biology and Evolution*. 9:1687–1698.

Rehan SM, Glastad KM, Lawson SP, Hunt BG. 2016. The genome and methylome of a subsocial small carpenter bee, *Ceratina calcarata*. *Genome Biology and Evolution*. 8:1401–1410.

Glastad KM, Hunt BG, Goodisman MAD. 2015. DNA methylation and chromatin organization in insects: insights from the ant *Camponotus floridanus*. *Genome Biology and Evolution*. 7:931–942.

Hunt BG, Glastad KM, Yi SV, Goodisman MAD. 2013. Patterning and regulatory associations of DNA methylation are mirrored by histone modifications in insects. *Genome Biology and Evolution*. 5:591–598.

L321 – How to interpret this extremely limited set of genes? Is this just random, or is this a statistically significant relationship between methylation and alternative exon expression?

We have updated our discussion to address this issue, as follows: “This suggests that a dynamic relationship between DNA methylation and splicing may not exist in *C. calcarata*. At minimum, it is clear from our data that changes in splicing corresponding to changes in DNA methylation are the exception rather than the rule. Changes in DNA methylation do not appear to consistently lead to changes in splicing, nor *vice versa*, at a level observable in our data set.”

Reviewers' comments:

Reviewer #2 (Remarks to the Author):

This is a re-review of a manuscript addressing the gene expression and methylation correlates of maternal care. In general, this is a well-done study involving a large amount of work and data, and the response to reviewers has significantly improved the nitty-gritty of the methods and interpretations (although please see a couple of questions below). In my view, there are three potential issues with the manuscript that should be addressed if possible. The first and third I brought up in my first review, and the second emerged somewhat from the reframing. First, it might matter that the bees do not commonly find themselves in a lack of care situation. That means that the differences between care and no-care situations are not necessarily concerted, evolved responses. Thus if the idea is that methylation and transcription are linked in the treatment responses, then that linkage may be intact in the care treatment but not necessarily in the no-care treatment. Then their observations could be attributed to a subtle role for methylation, or could be due to this lack of a concerted response. Perhaps they think this linkage may still be intact because all related species are raised in the no-care situation. If so, this might be helpful to state. Second, their message about the role of methylation is somewhat diluted in the text. On the one hand, they see little overlap between transcription and methylation and want to argue that the role is potentially nuanced and subtle. On the other hand, they argue that some differential methylation may be especially important because it overlaps with differentially used exons. Last, I'm still skeptical of the breadth of the broader implications of this study. It has a bit of a split personality about if it is addressing the functional basis of care/no care phenotypes versus a more general analysis of the overlap between expression and methylation. With respect to the former, the study is exploratory and the results are correlational and suggestive, and best viewed as hypotheses for future studies. With respect to the latter, it's somewhat interesting that they can find some differentially methylated regions, since I'm not aware of many (any?) insect studies with replicates that have found differential methylation between phenotypes.

And just a few of more minor questions:

Lines 159-162: Please include more detail about how many spanning reads were required to confidently call a differential exon usage.

Figure 1C: Are these one-sided or two-sided Mann Whitney U tests? And please include a justification for the choice if it is one-sided. Aggression, avoidance, and tolerance were measured. Aggression and avoidance are presented in the figure. What about tolerance? Was it not significant? I don't think this is in the text, either.

Lines 273-275: The authors say that the two different packages used to identify alternative splicing produced largely mutually exclusive lists of genes. This seems problematic, yet is not brought up again. Have they performed all of their alternative splicing analyses with both gene lists? Is there any reason to think that one list is better, maybe more or less conservative, than the other? I'm particularly interested in the splicing events in which they

found differential methylation. Do those overlap between the analyses? And if they don't, can they determine why not by looking more closely at the data?

Reviewer #3 (Remarks to the Author):

The authors have done a nice job addressing my extensive previous comments.

Reviewer #2 (Remarks to the Author):

This is a re-review of a manuscript addressing the gene expression and methylation correlates of maternal care. In general, this is a well-done study involving a large amount of work and data, and the response to reviewers has significantly improved the nitty-gritty of the methods and interpretations (although please see a couple of questions below). In my view, there are three potential issues with the manuscript that should be addressed if possible. The first and third I brought up in my first review, and the second emerged somewhat from the reframing.

1. First, it might matter that the bees do not commonly find themselves in a lack of care situation. That means that the differences between care and no-care situations are not necessarily concerted, evolved responses. Thus if the idea is that methylation and transcription are linked in the treatment responses, then that linkage may be intact in the care treatment but not necessarily in the no-care treatment. Then their observations could be attributed to a subtle role for methylation, or could be due to this lack of a concerted response. Perhaps they think this linkage may still be intact because all related species are raised in the no-care situation. If so, this might be helpful to state.

The potential for no-care situations in the wild is indeed an important consideration for appropriately interpreting the role of natural selection in shaping our results. We do not have a dataset to assess frequency of maternal death directly, but we have clarified in the discussion the importance of potential variation in maternal care in the wild, as follows: “Adaptive and non-adaptive explanations exist for these results. For example, divergent behaviors observed between bees reared with and without maternal care may simply reflect non-adaptive effects of a suboptimal developmental environment. However, plasticity in aggression may also be influenced by natural selection if maternal death occurs regularly in the wild or variation in maternal care is prevalent.”

2. Second, their message about the role of methylation is somewhat diluted in the text. On the one hand, they see little overlap between transcription and methylation and want to argue that the role is potentially nuanced and subtle. On the other hand, they argue that some differential methylation may be especially important because it overlaps with differentially used exons.

In our opinion, these are not mutually exclusive options, and we think this is an accurate description of our messaging on DNA methylation. Post-embryonic DNA methylation variation is clearly not a major transcriptional regulator in *C. calcarata* based upon our data, but on occasion DNA methylation exhibits localized associations with alternative splicing that may be important to developmental plasticity and such candidate loci are worthy of follow-up study.

3. Last, I’m still skeptical of the breadth of the broader implications of this study. It has a bit of a split personality about if it is addressing the functional basis of care/no care phenotypes versus a more general analysis of the overlap between expression and methylation. With respect to the former, the study is exploratory and the results are correlational and suggestive, and best viewed as hypotheses for future studies. With respect to the latter, it’s somewhat interesting that they can find some differentially methylated regions, since I’m not aware of many (any?) insect studies with replicates that

have found differential methylation between phenotypes.

As stated in the introduction, “The purpose of our study is twofold. First, we investigate the effects of variation in maternal care on offspring behavior and gene regulation, using the subsocial bee *C. calcarata* as a model. Second, we investigate whether stable differences in gene regulation observed between adults from distinct developmental environments are associated with, and potentially driven by, differences in DNA methylation.” There are indeed two main objectives of the study. We think these complementary aspects of our manuscript help to broaden its audience. With respect to the detection of differential DNA methylation, there are at least two other studies that robustly document differential DNA methylation in insects with biological replication, cited below and in our manuscript.

Glastad KM, Gokhale K, Liebig J, Goodisman MAD. 2016. The caste- and sex-specific DNA methylome of the termite *Zootermopsis nevadensis*. *Sci. Rep.* 6:37110. doi: 10.1038/srep37110.

Herb BR et al. 2012. Reversible switching between epigenetic states in honeybee behavioral subcastes. *Nat Neurosci.* 15:1371–1373. doi: 10.1038/nn.3218.

And just a few of more minor questions:

Lines 159-162: Please include more detail about how many spanning reads were required to confidently call a differential exon usage.

In our implementation of DEX-seq, no explicit coverage cutoffs were put in place. DEX-seq is implemented here (and generally) using a library normalization method (similar to DESeq2), dispersion (~variance) estimation, and then differential exon usage based on the variance in fit between the two sample types. That being said, among the exons that were deemed differentially utilized, the minimum per sample average number of spanning reads (across the 6 samples) was 3. Because exon size can vary widely and some interesting differentially utilized exons may be smaller, we did not want to set any firm thresholds for exons to be considered differentially utilized.

Figure 1C: Are these one-sided or two-sided Mann Whitney U tests? And please include a justification for the choice if it is one-sided. Aggression, avoidance, and tolerance were measured. Aggression and avoidance are presented in the figure. What about tolerance? Was it not significant? I don't think this is in the text, either.

The Mann Whitney U tests were two-sided and we have corrected the appropriate sentence in the text of the methods to clarify that we do not have tolerance data.

Lines 273-275: The authors say that the two different packages used to identify alternative splicing produced largely mutually exclusive lists of genes. This seems problematic, yet is not brought up again. Have they performed all of their alternative splicing analyses with both gene lists? Is there any reason to think that one list is better, maybe more or less conservative, than the other? I'm particularly interested in the splicing events in which they found differential methylation. Do those overlap between the analyses? And if they don't, can they determine why not by looking more closely at the data?

The two packages used to assess alternative splicing each use very different methods. rMATS analyses were conducted during revisions to assign more diverse types of alternative splicing than the differential exon usage assigned by DEXseq. Functionally, we're using primarily two mutually exclusive sets of reads to assess splicing with the different packages (exon reads for DEX-seq vs. junction-spanning reads with rMATS). We now clarify this issue in the methods, as follows. "rMATS is ideal for identifying a variety of alternative splicing event types, but its dependence on junction-spanning reads makes it liable to miss alternative splicing events that occur in low proportion. DEXseq, while only able to detect specific types of splicing events, leverages more total reads to yield a more thorough analysis of differential exon usage." Also, we already state in the the Results section, "Among the eight genes that were both differentially methylated and alternatively spliced in our data, three had a DMR directly overlapping a differentially utilized exon (Figure 4B-D, Figure S2) while none of the alternative splicing events discovered using rMATS overlapped directly with a DMR (Figure S3)."

Reviewer #3 (Remarks to the Author):

The authors have done a nice job addressing my extensive previous comments.

No revisions were requested by reviewer 3.

REVIEWERS' COMMENTS:

Reviewer #2 (Remarks to the Author):

The authors have addressed all of my comments.